# Technical Evaluation of the COBAS EGFR Semiquantitative Index (SQI) for Plasma cfDNA Testing in NSCLC Patients with EGFR Exon 19 Deletions

**DOI:** 10.3390/diagnostics11081319

**Published:** 2021-07-22

**Authors:** José Manuel González de Aledo-Castillo, Samira Serhir-Sgheiri, Neus Calbet-Llopart, Ainara Arcocha, Pedro Jares, Noemí Reguart, Joan Antón Puig-Butillé

**Affiliations:** 1Biochemistry and Molecular Genetics Department, Hospital Clínic, 08036 Barcelona, Spain; gonzalezde@clinic.cat; 2Molecular Biology CORE, Hospital Clínic, 08036 Barcelona, Spain; samira_serhir@hotmail.com (S.S.-S.); pjares@clinic.cat (P.J.); 3Melanoma Unit, Division of Oncology and Hematology, August Pi i Sunyer Biomedical Research Institute (IDIBAPS), 08036 Barcelona, Spain; calbet@clinic.cat; 4Medical Oncology Department, Hospital Clínic, 08036 Barcelona, Spain; aarcocha@clinic.cat (A.A.); nreguart@clinic.cat (N.R.); 5Thoracic Oncology Unit, Medical Oncology Department, Hospital Clínic, 08036 Barcelona, Spain; 6Pathology Department, Hospital Clínic, 08036 Barcelona, Spain

**Keywords:** EGFR, VAF, SQI, mutated copies/mL, NSCLC, cfDNA, cobas

## Abstract

The cobas^®^ EGFR Test provides a semiquantitative index (SQI) that reflects the proportion of mutated versus wild-type copies of the *EGFR* gene in plasma. The significance of SQI as an indirect measure of the variant allele frequency (VAF) or mutated copies/mL remains unclear. The aim of this study was to evaluate the correlation of SQI with the VAF and the number of mutated copies/mL obtained by a digital droplet PCR (ddPCR) test in NSCLC samples. The study included 118 plasma samples from a retrospective cohort of 25 stage IV adenocarcinoma patients with *EGFR* exon 19 deletions (Ex19Del), obtained before and during tyrosine kinase inhibitor (TKI) treatment. Both SQI and VAF and SQI and mutated copies/mL showed the same significant correlation (r^2^ = 0.79, *p* < 0.00001) across the whole study cohort. We found better correlation in samples collected at the baseline between SQI and VAF (r^2^ = 0.94, *p* < 0.00001) and SQI and mutated copies/mL (r^2^ = 0.97, *p* < 0.00001) compared to samples collected during TKI treatment: r^2^ = 0.76; *p* < 0.00001 for SQI and VAF and r^2^ = 0.75; *p* < 0.00001 for SQI and mutated copies/mL. The study indicates that SQI is a robust quantitative indirect measure of VAF and the number of mutated copies/mL in plasma from patients with an *EGFR* Ex19Del mutation. Further studies are desirable to assess the SQI cut-off values related to the clinical status of the patient.

## 1. Introduction

Molecular characterization of non-small cell lung cancer (NSCLC) has allowed a better classification of NSCLC tumors, permitting the introduction of personalized therapies that reduce toxicity and increase survival rates [1]. Recurrent genetic drivers in NSCLCs are mutations in the epidermal growth factor receptor (*EGFR*) gene. These mutations are present in 10 to 30% of patients [2], with deletions in exon 19 (Ex19Del) being the most common *EGFR* mutation [3]. *EGFR* tyrosine kinase inhibitors (TKIs) are currently the standard-of-care in first line therapy for patients with mutant *EGFR* NSCLC, with new generations of TKIs developed for patients with acquired resistance to the first and second generation TKIs [4]. This enhances the need for simplified strategies to measure the patient’s molecular response to TKI treatment to enable further treatment optimization and improved outcomes.

Tumor tissue analysis remains the gold standard to molecularly stratify advanced NSCLC patients to select optimal TKI treatment, but cell-free DNA (cfDNA) analysis is now recommended when the tissue biopsy is not available or insufficient [5]. Moreover, cfDNA from plasma is more informative than tissue analysis since it captures the heterogeneity of the tumor [6]. The analysis of cfDNA to molecularly stratify advanced NSCLC patients has proven useful in the clinical setting by our group, among others [7]. Beyond the qualitative information provided by such testing, the quantitative information of the variant allele frequency (VAF) has been proposed as a biomarker for monitoring treatment response [8,9], for minimal residual disease assessment (MRD) [10], and for relapse prediction in NSCLC patients [11].

Distinct approaches for cfDNA analysis exist such as real-time PCR (RT-PCR), beads, emulsion, amplification, and magnetics (BEAMing), droplet digital PCR (ddPCR), and next generation sequencing (NGS). These techniques differ both in terms of the sensitivity they achieve and their complexity [12]. Another technique is the commercial assay cobas^®^ EGFR Mutation Test v2 (cobas^®^ EGFR Test, Hoffman-La Roche Ltd., Basel, Switzerland). This test was approved by the FDA in 2016 as the first test for the identification of NSCLC patients harboring *EGFR* mutations in cfDNA from plasma [13] for TKI treatment selection. The cobas^®^ EGFR Test is a RT-PCR based method that interrogates 42 mutations located in exons 18, 19, 20, and 21 of the *EGFR* gene [14] and provides a semiquantitative index (SQI) for the *EGFR* mutation, which reflects the trend for the proportion of mutated versus wild-type copies of the *EGFR* gene in the cfDNA. However, the cobas^®^ EGFR Test is not approved as a quantitative test [15].

Robust and sensitive techniques are needed to translate liquid biopsy strategies to the clinical setting, since the threshold level of mutation corresponding to therapeutic implications is extremely low (VAF < 0.1%) [11,16]. The cobas^®^ EGFR Test has a detection limit of around 0.1–1% of the VAF [17]. In comparison, ddPCR techniques have a higher VAF sensitivity with a detection limit of 0.01–0.1% while also quantifying the absolute number of *EGFR* mutated copies (copies/mL) [18].

The correlation between SQI provided by the cobas^®^ EGFR Test and the number of *EGFR* mutated copies/mL and VAF has previously been explored with contradictory results. One study observed that SQI reflects the VAF of the *EGFR* mutant allele obtained by NGS methods and that dynamic changes in SQI reflect the tumor progression [19]. In contrast, Macías et al. [20] did not find a significant correlation between SQI and the number of *EGFR* mutated copies/mL obtained by ddPCR, concluding that the parameters are not interchangeable. One external quality assurance (EQA) program found a good correlation between SQI and VAF for *EGFR* Ex19Del, but showed low reproducibility of SQI when VAF was <1% [21]. Other studies have not found steady correlations between SQI and VAF for different *EGFR* mutations [22,23]. The reproducibility and correlation of SQI values to VAF and mutated copies/mL need to be validated in real patient material before using the data for the interpretation of clinical samples [24]. Further studies are therefore necessary to elucidate the value of SQI as a measure of VAF in *EGFR* mutations.

In this study, we evaluated the SQI parameter from the cobas^®^ EGFR Test as a measure for the number of mutated copies/mL and for the VAF of mutant *EGFR* allele in plasma in advanced NSCLC patients harboring an Ex19Del of *EGFR* gene.

## 2. Materials and Methods

### 2.1. Patients and Plasma Samples

The study included 118 cfDNA samples from plasma belonging to 25 stage IV adenocarcinoma patients harboring an Ex19Del of *EGFR* gene. The samples were collected at Hospital Clínic of Barcelona between June 2017 and May 2019. A sample was collected at the baseline before TKI treatment in 12 patients, and in the remaining 13 patients, all samples were collected exclusively over the clinical follow-up during TKI treatment. Clinical data such as gender, age, tumor histology, molecular status in tumor, and disease stage were obtained from the patients’ medical records.

The patients belong to a prospective observational study approved by the Hospital Clínic Ethics Committee (approval registration number HCB/2016/0889). The study was conducted in accordance with the precepts of the Code of Ethics of The World Medical Association (Declaration of Helsinki). Written informed consent was obtained from all patients.

### 2.2. Extraction of cfDNA from Plasma Samples

Peripheral whole blood was collected from each subject in a 5 mL EDTA-K2 tube. After 15 to 20 min at rest in an upright position at room temperature, samples were centrifuged at 1600× *g* for 10 min to collect 2 mL of plasma, which was transferred to a sterile tube. After a second centrifugation at 16,000× *g* for 10 min, plasma samples were stored at −20 C. The entire procedure was completed within three hours of blood extraction. The cfDNA was isolated using the cobas^®^ cfDNA Sample EGFR Preparation Kit as per the manufacturer’s instructions.

The cfDNA quantification and quality assessment were performed using a Qubit^®^ 2.0 Fluorometer (Life Technologies, Carlsbad, CA, USA) with the Qubit^®^ dsDNA HS Assay Kit, and an Agilent 2100 Bioanalyzer (Agilent Technologies, Santa Clara, CA, USA) with the Agilent High Sensitivity DNA Assay respectively.

### 2.3. Molecular Characterization of EGFR in cfDNA from Plasma

Analysis of EGFR mutations in cfDNA was first assessed with the cobas^®^ EGFR Test (Hoffman-La Roche Ltd., Basel, Switzerland), following the manufacturer’s recommendations, which provided a SQI value along with the detected EGFR mutation. The limit of detection for *EGFR* Ex19Del declared by the manufacturer is 75 mutated copies/mL.

A second analysis of the EGFR mutations was then performed using a ddPCR^TM^ EGFR exon 19 Deletion Screening Kit Assay (ddPCR^TM^ EGFR Test, Bio-Rad Laboratories, Hercules, CA, USA). The ddPCR^TM^ EGFR Test detects 15 deletion mutations in exon 19 of the *EGFR* gene. The ddPCR was carried out in a reaction volume of 20 µL on a QX200TM Droplet Digital^TM^ PCR System (Bio-Rad). The 20 µL PCR mix contained 10 µL of 2× ddPCR Supermix for probes (No dUTP), 1 µL of 20× of the corresponding probe, 1 µL of ECOR1 enzyme, 1 µL of water, and 7 µL of plasma cfDNA. Droplets were generated by the QX200^TM^ droplet generator and PCR was performed using a C1000^TM^ Touch thermal cycler (Bio-Rad). Reading and analysis was executed using Quantasoft^TM^ software (Bio-Rad). The results were reported as wild-type and the number of mutated copies/mL of plasma and the VAF of the *EGFR* mutation was then calculated as ((mutated copies/mL)/(total copies/mL)) × 100. The criteria to determine if the assay worked properly were: number of accepted droplets ≥13,000, good separation between the different clusters, and at least 50 copies/reaction. The limit of detection reported by the manufacturer is 0.5% of VAF.

### 2.4. Statistical Analysis

The agreement between the two methods was measured by calculating the Kappa coefficient. Spearman rank correlation was used to analyze the associations between SQI and VAF and SQI and the number of mutated copies/mL. Results were considered statistically significant with a *p*-value < 0.05. All statistical analysis was performed using STATA/IC software version 13.1 (StataCorp LLC, College Station, TX, USA).

## 3. Results

### 3.1. Study Cohort

The 25 stage IV adenocarcinoma patients consisted of 18 females and seven males, harboring a deletion in exon 19 of the *EGFR* gene. We selected the patients included in the study based on a positive result obtained by the initial cobas^®^ EGFR Test in the first sample tested.The mean age at first sample collection was 68.2 ± 13.6 years old. Overall, we analyzed 118 plasma samples ranging from 1 to 20 (Median = 3) per patient. A plasma sample was collected at baseline before TKI treatment in 12 patients (N = 12 samples), and only during follow-up in 13 patients (N = 85 samples). Nine of the 12 patients analyzed at the baseline also had at least one sample analyzed in follow-up (N = 21). The molecular status of *EGFR* in tumor tissue was available in 60% (15/25) of patients. In this subset of patients, five different *EGFR* mutations were found: four deletions and one insertion (Table 1).

The cobas^®^ EGFR Test and the ddPCR^TM^ EGFR Test differ in which exon 19 deletions are evaluated. Thirteen mutations are common to both tests, while the cobas^®^ EGFR Test includes a further 16 mutations not included in the ddPCR^TM^ EGFR Test. The ddPCR test also analyzes two mutations not included in the cobas^®^ EGFR Test. Based on the characterization of tumor tissue, the five EGFR mutations could be detected by the cobas^®^ EGFR Test while two patients harbored an *EGFR* Ex19Del included in the cobas^®^ EGFR Test but not included in the ddPCR^TM^ EGFR exon 19 deletion screening kit. However, the ddPCR analysis resulted in positive droplets in samples belonging to these patients. In the patient who harbored the c.2337_2255insT *EGFR* mutation, positive droplets were observed in both samples analyzed. In the other patient, with the c.2240_2251del12 *EGFR* mutation, positive droplets were found in two out of four samples analyzed.

### 3.2. Agreement between Cobas^®^ EGFR Test and the ddPCR^TM^ EGFR Test

First, we evaluated the agreement between the cobas^®^ EGFR Test and the ddPCR Test and found discrepancies in the positive results obtained by each method. A total of 72.0% (85/118) of samples presented an *EGFR* Ex19Del positive result by at least one method. We detected a positive result by the cobas^®^ EGFR Test in 76 samples but in 26.3% (20/76) of these, the *EGFR* mutation was not detected by ddPCR. In all of these cases, the SQI value reported was below 12. On the other hand, we found a positive Ex19Del *EGFR* result by ddPCR in 65 samples, of which the *EGFR* mutation was not detected by the cobas^®^ EGFR Test in 13.8% (9/65) of cases. In all these samples, the number of mutated copies was <75 copies/mL. All samples that had a positive result by the ddPCR^TM^ EGFR Test in which the mutation was not detected by the cobas^®^ EGFR Test were obtained during follow-up under TKI treatment.

Next, we analyzed the samples regarding Ex19Del *EGFR* status depending on TKI treatment status. In the subset of samples obtained at the baseline before TKI treatment, an Ex19Del *EGFR* mutation was detected in 100% (12/12) of patients by the cobas^®^ EGFR Test and in 75.0% (9/12) by the ddPCR^TM^ EGFR Test. In samples drawn after the TKI treatment was initiated, a mutation was found in 60.4% (64/106) of samples by the cobas^®^ EGFR Test and in 52.8% (56/106) by ddPCR.

The agreement between both methods for samples at the baseline before TKI treatment was 75%, and for the follow-up samples, it was 75.5% with a Kappa coefficient of 0.50 (*p* < 0.00001). The overall agreement between all samples was 75.4% with a Kappa coefficient of 0.49 (*p* < 0.00001).

The median SQI of Ex19Del EGFR mutation was significantly lower in those samples where there was no agreement between the two methods (8.3 vs. 12.1, *p* < 0.00001) as well as VAF (0.2 vs. 0.8, *p* = 0.002) and the number of mutated copies/mL (11.7 vs. 49.0, *p* = 0.02).

We also evaluated the agreement between the cobas^®^ EGFR Test and the ddPCR^TM^ EGFR Test in the subset of samples where tissue analysis was available (N = 15 patients, 79 samples) and obtained concordant results to those of the whole study cohort. The percentage of samples that presented a positive result by at least one method was slightly higher (73.4%, 58/79), and the discrepancies between methods were slightly lower (21.2% for the mutations detected by the cobas^®^ EGFR Test and not by ddPCR, and 12.8% the other way around).

An Ex19Del *EGFR* mutation was detected in 100% (6/6) of patients by the cobas^®^ EGFR Test and in 66.7% (4/6) by the ddPCR^TM^ EGFR Test in samples obtained before TKI treatment and in 63.0% (46/73) of samples by the cobas^®^ EGFR Test and in 58.9% (43/73) by ddPCR in samples drawn after the TKI treatment was initiated. The agreement for samples at the baseline before TKI treatment was 66.7% and for follow-up samples 79.5% with a Kappa coefficient of 0.57 (*p* < 0.00001). The overall agreement between all samples was 78.5% with a Kappa coefficient of 0.54 (*p* < 0.00001).

As in the whole study cohort, the median SQI of Ex19Del EGFR mutation was significantly lower in those samples where there was no agreement between the two methods (9.0 vs. 12.1, *p* < 0.001). However, VAF (0.3 vs. 0.7, *p* = 0.07) and the number of mutated copies/mL (17.5 vs. 49.4, *p* = 0.11) were also lower in samples with no agreement, but these differences were not significant.

### 3.3. The SQI from Cobas^®^ EGFR Test Correlates with the VAF and the Number of Mutated Copies/mL from ddPCR^TM^ EGFR Test

In patients with an Ex19Del mutation detected by the cobas^®^ EGFR Test, the median SQI value was 11.1 (IQR = 4.3), ranging from 6.0 to 22.5. We observed higher SQI values for samples at the baseline before TKI treatment than for those taken during follow-up under TKI treatment (Median = 12.4 vs. 11.0, *p* = 0.19). In patients with an Ex19Del mutation detected by ddPCR, the median VAF was 0.62 (IQR = 2.5), ranging from 0.04 to 39.3; and the median mutated copies/mL was 34.0 (IQR = 123.3), ranging from 9.2 to 42798. Samples obtained at the baseline before TKI treatment presented higher VAF values (median = 1.6 vs. 0.4, *p* = 0.02) and mutated copies/mL (Median = 109.5 vs. 25.4, *p* = 0.03) than the samples collected during follow-up.

We evaluated the correlation between the SQI and the VAF value in the whole set of samples and found a statistically significant correlation between both parameters (r^2^ = 0.79, *p* < 0.00001). Next, we evaluated the samples based on TKI treatment. Samples collected before TKI treatment showed a better correlation (r^2^ = 0.94, *p* < 0.00001) than the samples obtained after TKI treatment (r^2^ = 0.76, *p* < 0.00001).

We obtained similar results comparing SQI to the number mutated copies/mL. For the whole study cohort, we found a statistically significant correlation between SQI and the mutated copies/mL (r^2^ = 0.79, *p* < 0.00001). A higher correlation was detected for samples before TKI treatment (r^2^ = 0.97, *p* < 0.00001) than for samples obtained after TKI treatment (r^2^ = 0.75, *p* < 0.00001) (Figure 1).

In the subset of patients in which tissue analysis was available, the correlations for SQI and VAF for both the whole set of samples and for samples from patients under TKI treatment were practically the same as for the whole study cohort (r^2^ = 0.78, *p* < 0.00001 and r^2^ = 0.8., *p* < 0.00001, respectively). However, in samples collected before TKI treatment was initiated, the correlation was better in patients where the tissue result was known (r^2^ = 0.99, *p* = 0.0003). The same trend was observed for the SQI and mutated copies/mL, where the correlation for the whole set of samples was r^2^ = 0.75 (*p* < 0.00001), for samples before TKI treatment it was r^2^ = 0.99 (*p* = 0.0003), and for samples obtained after TKI treatment, it was r^2^ = 0.72 (*p* < 0.00001).

### 3.4. Example of Correlation between SQI and VAF and SQI and Mutated Copies/mL in a Patient with Longitudinal Follow-Up

One patient included in the study had a long follow-up with frequent monthly cfDNA sample analysis (ID15). She was monitored with the cobas^®^ EGFR Test every month from three months after initial diagnosis (sample 1) until follow-up was lost. At the time of the first sample included in this study, the patient had already been under TKI treatment for three months. The agreement between the result of the cobas^®^ EGFR Test and the ddPCR^TM^ EGFR Test was 85.0% (17/20). Two out of the three discordant results corresponded to two samples that tested positive for the ddPCR method, but not for the cobas^®^ EGFR Test. These two samples showed low VAF values (0.23% and 0.15%) and 23.4 and 10.4 mutated copies/mL. The sample that tested positive for the cobas^®^ EGFR Test, but not for the ddPCR method had a SQI value of 9.0. The correlation between the SQIs and VAF and SQI and mutated copies/mL was r^2^ = 0.79 (*p* < 0.00001) (Figure 2a). When representing the SQI value and the natural log (ln) of the mutated copies/mL throughout the chronological monitoring process of this patient, we can see how both SQI and the mutated copies/mL were zero or very low during the first 13 monitoring analyses. However, in sample 14, a small amount of mutated copies/mL was detected. In sample 15, the SQI appeared and in sample 16, both parameters reached their highest level thus far. In sample 17, the resistance mutation T790M was detected along with the sensitizing mutation Ex19Del. Between sample 17 and sample 18, treatment with Osimertinib, a third generation TKI, began, and both SQI and mutated copies/mL began to decline until sample 20, when the resistance mutation T790M disappeared and SQI and mutated copies/mL returned to low levels (Figure 2b).

## 4. Discussion

In this study, we compared the qualitative and quantitative performance of the cobas^®^ EGFR Test and the ddPCR^TM^ EGFR Test for the detection of deletions in exon 19 in the EGFR gene in 118 samples from 25 NSCLC patients. The overall agreement between both tests was 79%, which was slightly lower compared to other qualitative studies [25]. The lower number of probes in the ddPCR^TM^ EGFR Test compared to the cobas^®^ EGFR Test as well as technical factors such as the differential performance of these probes in samples with low mutated copies/mL might have affected our results.

Serial molecular analysis of cfDNA from plasma during treatment may have potential clinical utility, but the periodicity of longitudinal blood sampling is not yet established [26]. In our study, for follow-up samples during treatment, we obtained several negative results using one method that gave positive results in the other using the Cobas^®^ EGFR and the ddPCR^TM^ EGFR Test, regardless of their analytical sensitivity. The false negative results seen for the cobas^®^ EGFR Test, involved samples with less than 75 mutated copies/mL, which is the limit of detection declared by the manufacturer, and agrees with other published papers [24]. In the case of the false negative results seen with the ddPCR method, the EQA study [21] has shown that SQI values below 12 correlate to VAFs below 0.5%, which is the limit of detection reported by the manufacturer for this method. In our study, all samples that gave a false negative result on the ddPCR method had a SQI value below 12. This emphasizes the need for higher sensitivity methods like ddPCR to test samples collected during TKI treatment, where the response to the treatment results in a decrease in the tumor burden and consequently lower levels of mutation.

There is no consensus on how to act in front of a negative result during monitoring analysis. Clinical guidelines recommend a tissue biopsy to assess the resistance mutation T790M if plasma testing is negative due to the high risk of false negative results [26,27], but no such recommendations have been set for sensitizing mutations. One way to approach this issue would be to reanalyze the negative sample through an alternative method where possible. Another approach would be to repeat the blood extraction and plasma analysis in a defined period, following the example of serum tumor marker analysis, in which retesting is recommended at 3–4 weeks, or at least a period longer than the tumor marker’s plasma half-life, which is 15–20 days for most of them [28,29], or, as recommended for radiological diagnosis, every 6–12 weeks for advanced NSCLC [30] or 6–12 months for stage III NSCLC [31]. As liquid biopsy is rapidly gaining more relevance in clinical practice, the need for standardized protocols and algorithms regarding its usefulness as a treatment monitoring tool has become an important issue that must soon be addressed by clinical guidelines.

Both the cobas^®^ EGFR Test [24,32] and ddPCR tests [33,34] have proven useful in routine clinical practice to molecularly stratify NSCLC patients. Fast and robust tests like the cobas^®^ EGFR Test are useful as an initial approach for molecular characterization. However, in terms of quantitative outcomes, the significance of the SQI value and its role as a reliable measure of VAF remains unclear. This is a crucial point when the cobas^®^ EGFR Test in cfDNA is used for quantitative rather than qualitative purposes such as for evaluating minimal residual disease or for monitoring TKI treatment response. The EQA study showed that the SQI value presents high imprecision among different EGFR mutations, especially at low VAF values [21]. In a previous study, we observed a lack of correlation between the SQI value with the cfDNA concentration or the stage of the disease, and a moderate reproducibility that differed between distinct mutations [7]. However, in this study, we focused on the most recurrent *EGFR* alterations, and found a strong correlation between the SQI value and VAF and SQI and the number of mutated copies/mL, indicating that the SQI accurately reflects the VAF kinetics in NSCLC patients harboring deletions in exon 19 in EGFR. The correlation was stronger in samples collected before the initiation of TKI treatment compared to samples obtained during treatment, most probably because the circulating tumor DNA (ctDNA) levels significantly decreased with TKI treatment [35]. Notably, our study and the EQA study found a similar correlation between SQI and the VAF for deletions in exon 19. However, the EQA study was carried out with reference standards spiked to normal human plasma and the correlation was assessed only as part of a sensitivity study evaluation, while our results were obtained from samples collected in a real clinical setting, which enhances the translational meaning of our work, since the results were validated under the standardized conditions of the clinical laboratory. Therefore, based on our results, the SQI value is a robust quantitative measure that has the potential to provide useful information of the mutation load of the tumor. We saw in the longitudinal example shown in the study that the SQI and the mutated copies/mL showed similar kinetics in the patient’s follow-up, with low or undetectable values during treatment, higher levels when resistance was acquired, and again a descent in both parameters as the patient responded to new treatment. However, while cut-off values for VAF to predict progression-free survival were calculated both for activating *EGFR* (act-*EGFR*) mutations and for the ratio T790M/act-*EGFR* [8], further studies are needed to calculate SQI cut-offs with clinical relevance for the management of NSCLC patients, especially to detect TKI response failure or MDR.

When introducing a new test in a clinical laboratory, it must be verified or validated to fulfill international standard criteria [36]. For qualitative tests, it is important to calculate the standard measures of diagnostic accuracy, like sensitivity, specificity, and accuracy. False negative results can result in underdiagnosis, while false positives can result in inappropriate treatment, unnecessary tests, and anxiety for the patient [37]. It is worth noting that in our study, samples from two patients harboring EGFR mutations not available in the ddPCR^TM^ EGFR Test resulted in positive droplets. In samples with positive results with extremely low fractions of mutant DNA, the possibility of experimental artifacts cannot be completely ruled out. Non-specific annealing of PCR primers could result in a false positive result when the concentration of the wild-type template is much higher than the mutant template [38]. One of the false positive results in our study showed a very low amount of mutated copies/mL compared to the wild-type copies/mL. In the other confirmed false positive case, the proportion of mutated copies/mL vs. wild-type copies/mL was not low, which suggests another origin of the false positive result. In these cases, it is most likely due to cross-reactivity with the probes which are designed for mutations within the same region, as has been previously described in ddPCR methods [39]. These findings illustrate the importance of technical factors such as the specificity of designed probes in ddPCR that need to be evaluated prior to their use in the clinical setting to avoid potential false positive results.

One limitation of our study is the lack of molecular results obtained by tissue biopsy in 40% of the patients. This means that we cannot confirm that the mutations present are correctly covered by the cfDNA detection kits used in the study. However, since all patients without tissue biopsy returned positive results for both cfDNA detection kits used in the study, it is fair to assume that this should not significantly affect the results or the conclusions drawn. Due to the number of available samples, we focused on exon 19 deletions. Differences in SQI among distinct *EGFR* mutations [7,21] mean that our results cannot be extrapolated to other less common *EGFR* mutations such as the p.L858R *EGFR* mutation.

In conclusion, this study indicates that SQI strongly correlates with VAF and to the number of mutated copies/mL in patients with exon 19 deletions in the *EGFR* gene, highlighting that SQI is a robust quantitative measure whose magnitude could have clinical impact as confirmed in the longitudinal study of a patient. Additional studies are needed to assess the clinical relevance of SQI cut-off values for the management of NSCLC patients.

## Figures and Tables

**Figure 1 diagnostics-11-01319-f001:**
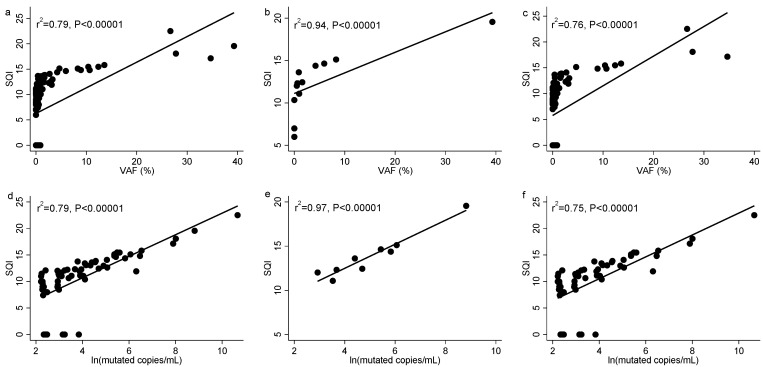
Correlation analysis between (**a**) SQI and VAF in all samples, (**b**) SQI and VAF in samples at the baseline before TKI treatment, (**c**) SQI and VAF in the samples after TKI treatment, (**d**) SQI and ln (mutated copies/mL) in all samples, (**e**) SQI and ln (mutated copies/mL) in samples at the baseline before TKI treatment, and (**f**) SQI and ln (mutated copies/mL) in samples after TKI treatment. Mutated copies/mL were transformed to their natural logarithm to ease graphical representation.

**Figure 2 diagnostics-11-01319-f002:**
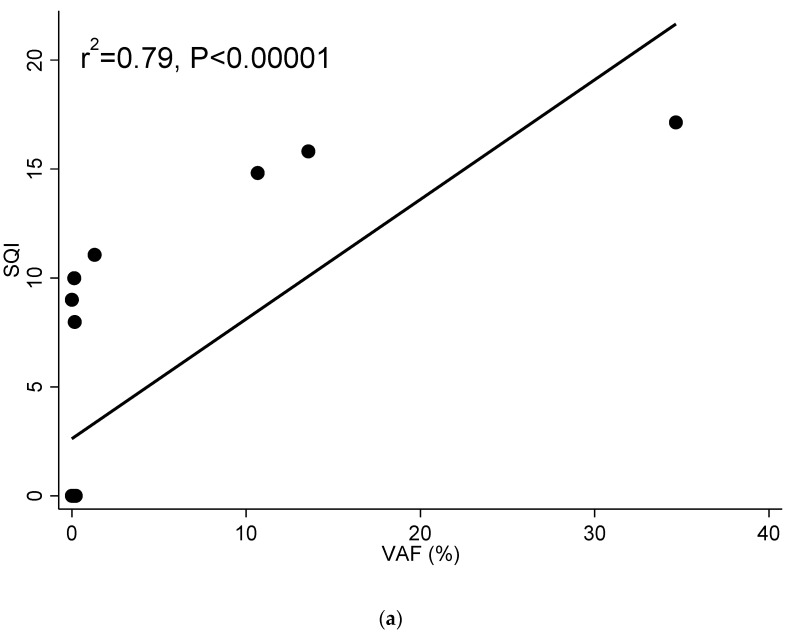
(**a**) Correlation between SQI and VAF in patient ID15. (**b**) Chronological monitoring of SQI and ln(mutated copies/mL) in patient ID 15. Samples in chronological order with one month between each sample. Guides to the eye are included to indicate the trend in the values as resistance mutation was found and treatment adjusted.

**Table 1 diagnostics-11-01319-t001:** Characteristics of lung cancer patients.

ID	Age	Sex	EGFR Status by Tissue Analyses	Number of Samples	Baseline Samples	Follow-Up Samples	Mutation Included in Cobas EGFR Test	Mutation Included in ddPCR Assay
ID01	91	F	NA	1	1	0	NA	NA
ID02	73	M	c.2235_2249del15	1	1	0	Yes	Yes
ID03	53	M	c.2240_2251del12	4	1	3	Yes	No
ID04	40	M	c.2236_2250del15	5	1	4	Yes	Yes
ID05	56	F	NA	3	1	2	NA	NA
ID06	70	M	NA	2	1	1	NA	NA
ID07	77	M	NA	3	1	2	NA	NA
ID08	87	M	c.2236_2250del15	2	1	1	Yes	Yes
ID09	72	F	c.2235_2249del15	6	1	5	Yes	Yes
ID10	87	F	c.2236_2250del15	2	1	1	Yes	Yes
ID11	70	F	NA	1	1	0	NA	NA
ID12	79	F	NA	3	1	2	NA	NA
ID13	61	F	c.2235_2249del15	2	0	2	Yes	Yes
ID14	81	F	NA	4	0	4	NA	NA
ID15	69	F	c.2240_2257del18	20	0	20	Yes	Yes
ID16	56	F	c.2337_2255insT	2	0	2	Yes	No
ID17	74	F	c.2235_2249del15	1	0	1	Yes	Yes
ID18	83	F	NA	8	0	8	NA	NA
ID19	67	F	NA	2	0	2	NA	NA
ID20	67	M	NA	12	0	12	NA	NA
ID21	80	F	c.2235_2249del15	1	0	1	Yes	Yes
ID22	45	F	c.2235_2249del15	11	0	11	Yes	Yes
ID23	49	F	c.2236_2250del15	5	0	5	Yes	Yes
ID24	58	F	c.2240_2254del15	2	0	2	Yes	Yes
ID25	61	F	c.2236_2250del15	15	0	15	Yes	Yes

ID: Identification; F: Female; M: Male; NA: Information not available.

## Data Availability

The datasets generated during and/or analyzed during the current study are available from the corresponding author on reasonable request.

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
