# Peer review of "Technical Evaluation of the COBAS EGFR Semiquantitative Index (SQI) for Plasma cfDNA Testing in NSCLC Patients with EGFR Exon 19 Deletions"

_diagnostics, 2021, doi:10.3390/diagnostics11081319_

Round 1

Reviewer 1 Report

In the present study José Manuel González de Aledo-Castillo et al evaluated the correlation of SQI with the VAF and the number of mutated copies/mL obtained by a digital droplet PCR (ddPCR) test in 79 plasma samples from 15 stage IV adenocarcinoma patients obtained before and during tyrosine kinase inhibitor (TKI) treatment. EGFR mutations in cfDNA was assessed with the cobas® EGFR Test and with ddPCR EGFR exon 19 deletion screening kit assay.

This study have several major revision that should be addressed.

Major Compulsory Revisions:

  1. It is not clear the purpose of this research. Cobas analysis is FDA approved by using SQI why the authors need to correlate SQI with VAF??
  2. Concordance between cobas® EGFR Test and the ddPCRTM EGFR Test. It is not clear for me how the authors compare to different kits/methods that measuring different mutations. Only thirteen mutations are common to both tests.
  3. There is limited number of patients.

Author Response

We thank the reviewers for their kind and supportive comments and suggestions and for the opportunity to clarify elements of the manuscript to improve and strengthen the results. Here we respond individually to their queries.

Reviewer 1

  1. It is not clear the purpose of this research. Cobas analysis is FDA approved by using SQI why the authors need to correlate SQI with VAF??

Reply: As you have mentioned, the cobas EGFR Test is an FDA approved test for the detection of exon 19 deletions or exon 21 (L858R) substitution mutations in the EGFR gene to identify patients with metastatic NSCLC eligible for treatment with TKIs. However, the SQI, which is displayed along with the qualitative result and it´s supposed to represent the amount of the target mutation detected, has not been validated yet as a biomarker to be used in routine clinical practice. The VAF has been proposed as biomarker of the tumour burden and for monitoring the disease evolution. Correlating both parameters, the SQI and the VAF, allows us to evaluate the feasibility of SQI as a surrogate of the VAF in patients which follow-up is made by the Cobas Test.

Moreover, an external quality assurance (EQA)[1] program has shown different grades of correlation of SQI and VAF in spiked samples depending on the EGFR mutation. We want to corroborate the results obtained by this EQA program in a real world clinical setting.

The introduction has been modified to clarify this point.

  1. Concordance between cobas® EGFR Test and the ddPCRTM EGFR Test. It is not clear for me how the authors compare to different kits/methods that measuring different mutations. Only thirteen mutations are common to both tests.

Reply: Thank you for your comment. As you correctly state, there are only thirteen common mutations between both tests. However, we have seen that both methods are capable to detect all mutations harboured by the subset of patients where the tissue result is known, even the two mutations not included in the ddPCR kit. In the subset of patients in which the mutation in tissue is not known, at least one positive analysis has been obtained per patient by both methods. Thus, we assume that all mutations included in the study can be detected by both methods. The agreement between both tests could help us to assess the interchangeability of these tests in the clinical practice.

We have modified the discussion to clarify this point.

  1. There is limited number of patients.

Reply: We are aware of this limitation. This is a preliminary study to assess the technical correlation between SQI and VAF. Although the number of patients is low, we think that the number of samples is valid for a technical evaluation. However, following your recommendation, we have added ten additional patients to the study, resulting in 39 more samples. The limitation of this addition is that we don´t have knowledge on the exact mutation status from a tissue biopsy, but since all ten additional patients gave positive results in both ddPCR and COBAS, it is fair to assume their mutation is covered by the tests. These extra samples allow us to strengthen our results and conclusions. Methods, results and discussion have been modified regarding this patient inclusion.

Reviewer 2 Report

This manuscript reports on their results comparing the cobas and ddPCR testing methods on the EGFR mutations in the cfDNA of NSCLC patients. Their results suggest the SQI is strongly correlated to the ddPCR results. The manuscript is well organized in general. A few points need to be addressed.

  1. One of the main limitations is the small patient number (15). This small number significantly limits the impact of this work. In particular, the sample numbers in Fig. 1b&e are too small (6&4). Why only 4 in Fig. 1e? The small sample pool before TKI treatment leads to the inclusive comparison in Line 250.
  2. Another limitation lies in the fact that no cut-off value of SQI is provided, which compromise its clinical significance.
  3. Line 119, is the criteria “>13,000” or “13.000”
  4. How many mutations were present in these 15 patients (4 del+ 1 inst?)? Can all these mutations be detected by the two methods? Surely, the mutations in the tissues do not necessarily agree with those in the cfDNA. But this is a critical piece of info which can help assess the accordance of the two methods. The info is in the table please also summarize it in the text.
  5. Can authors explain why only patient ID08 was followed up for so many times? It would also be interesting to see the comparison of the longitudinal testing from the same patient while receiving treatment.
  6. Line 166, please check the number 4/6. It appears to be 5/6 in the table.
  7. Can authors discuss false-positive and false negative problems of both methods?
  8. Can the authors explain why the PCR used for cobas does not give the absolute copy numbers per ml? Line 115-116 indicates that the copies were reported by the cobas testing as well.

Author Response

We thank the reviewers for their kind and supportive comments and suggestions and for the opportunity to clarify elements of the manuscript to improve and strengthen the results. Here we respond individually to their queries.

Reviewer 2

  1. One of the main limitations is the small patient number (15). This small number significantly limits the impact of this work. In particular, the sample numbers in Fig. 1b&e are too small (6&4). Why only 4 in Fig. 1e? The small sample pool before TKI treatment leads to the inclusive comparison in Line 250.

Reply: We are aware of this limitation. This is a preliminary study to assess the technical correlation between SQI and VAF. Although the number of patients is low, we think that the number of samples is valid for a technical evaluation. However, following your recommendation, we have added ten additional patients to the study, which means 39 more samples. The limitation of this addition is that we don´t have knowledge on the exact mutation status from a tissue biopsy, but since all ten additional patients gave positive results in both ddPCR and COBAS, it is fair to assume their mutation is covered by the tests. Methods, results and discussion have been modified regarding this patient inclusion.

Regarding the fact that Figure 1e shows fewer patients than Figure 1b, this is due to the limitation of representing the mutated copies/mL logarithmically, which means that samples with zero mutated copies/mL cannot be represented. We have added this information in the figure.

Concerning your last point of the sample pool before TKI treatment, with the addition of more patients, we have increased the sample size of this group, obtaining a practically identical result, which further strengthens our conclusions.

  1. Another limitation lies in the fact that no cut-off value of SQI is provided, which compromise its clinical significance.

Reply: Thank you for your comment. Establishing a cut-off value is crucial for the incorporation of these biomarkers to clinical practice. However, the SQI is a novel parameter and preliminary studies such as ours are first needed to assess the correlation with known biomarkers such as VAF. Once this correlation is fully elucidated, studies evaluating the cut-off value of clinical significance need to be performed with greater sample sizes. We have added a sentence in the discussion to address this matter.

  1. Line 119, is the criteria “>13,000” or “13.000”

Reply: The criteria is > 13,000 [thirteen thousand]. The number of droplets generated varies depending on the sample. If less than 13,000 droplets are generated, the results are not reliable. We have updated the manuscript to indicate the correct number.

  1. How many mutations were present in these 15 patients (4 del+ 1 inst?)? Can all these mutations be detected by the two methods? Surely, the mutations in the tissues do not necessarily agree with those in the cfDNA. But this is a critical piece of info which can help assess the accordance of the two methods. The info is in the table please also summarize it in the text.

Reply: The number of different mutations found, as you have mentioned, are 4 deletions and 1 insertion. All of them can be detected by the Cobas method, but two of them, one deletion and the insertion theoretically should not be detected by the ddPCR method, but as we have seen, they have been detected. These might be due to poor specificity in the probes, which can test positive in samples with similar mutations to the ones that the probe was initially designed. This point is further discussed in response 7. We have modified the discussion of the manuscript to directly address this point.

Regarding the second part of your question, the mutation that is present in the tissue are the same as the ones found in plasma. In patients where tissue sample is not available or difficult to obtain, a plasma sample is recommended to molecularly stratify patients[2]. Moreover, cfDNA from plasma is more informative than the tissue as it can capture the heterogeneity of the tumour[3]. We have included this information in the introduction.

As you have pointed out, it is important to know the exact mutation in tissue to be able to calculate the diagnostic accuracy and the agreement of the methods in evaluation. We have this information in most of the patients included in the study. In the new subset of patients, this information is not available. However, in this group of patients, at least one positive analysis has been obtained per patient by both methods. Thus, we assume that all mutations included in the study can be detected by both methods. We have added this information in the manuscript to clarify this point.

  1. Can authors explain why only patient ID08 was followed up for so many times? It would also be interesting to see the comparison of the longitudinal testing from the same patient while receiving treatment.

Reply: Patient ID08 (now, ID15 after the addition of new patients) has been followed every month after diagnosis until she died. Compared to other patients she had a particularly long follow-up period resulting in many samples. The longitudinal testing of this patient shows a good agreement between both methods as well as good correlation between SQI and VAF and SQI and mutated copies/mL in positive samples. Following the reviewer’s suggestion, we have expanded the analysis of this patient in the manuscript to show the dynamics of the VAF and SQI parameters throughout the patient’s treatment, resistance generation and change in treatment. The results further support the clinical relevance of these parameters in monitoring patients on treatment with these techniques. This analysis as well as two figures supporting it have been added to the text.

  1. Line 166, please check the number 4/6. It appears to be 5/6 in the table.

Reply: The numbers have been updated with the new sample size and now reads 9/12 patients. In the original manuscript, in samples before TKI treatment, only 4/6 patients tested positive with the ddPCR test. In the table it is not mentioned the number of patients that tested positive by each method, but if the probe was included in the kit.

  1. Can authors discuss false-positive and false negative problems of both methods?

Reply: The analysis of false positive and false negative results have to be performed with the subset of patients where the information of the tissue sample was available. The Cobas test did not present any false positive result, as all 5 different mutations found in tissue were available on the plasma kit. Regarding the false negative results (samples in which a positive result was obtained by the ddPCR but tested negative by Cobas), they are samples with less than 75 mutated copies/mL, which is the limit of detection declared by the manufacturer[4], and agree with other published papers[5].

Two of the mutations present in tissue but not included in the ddPCR kit were detected by the ddPCR method. In samples with positive results with extremely low fractions of mutant DNA, the possibility of experimental artefacts cannot be completely ruled out. Non-specific annealing of PCR primers could result in a false positive result when the concentration of wild-type template is much higher than mutant template[6]. One of the false positive results in our study showed a low amount of mutated copies/mL compared to the wild-type copies/mL. In the other confirmed false positive case, the proportion of mutated copies/mL vs. wild-type copies/mL was not low, which suggests another origin of the false positive result. In this cases, it is most likely due to cross-reactivity with the probes which are designed for mutations within the same region, as has been previously described in ddPCR methods[7].

The false-negative results in ddPCR may be due to low cfDNA input[8]. In the case of our study, cfDNA concentration was not significantly different between samples with false-negative results for ddPCR and the rest of samples of the study. Another hypothesis is the mutation load. The EQA study have shown that SQI values below 12 correlate with VAF below 0.5%, which is the limit of detection that the manufacturer has reported for this method. In our study, all samples that reported a false negative result on the ddPCR method had an SQI value below 12.

We have included a discussion of this causes of false positives within the manuscript.

  1. Can the authors explain why the PCR used for cobas does not give the absolute copy numbers per ml? Line 115-116 indicates that the copies were reported by the cobas testing as well.

Reply: The Cobas test is a qualitative PCR test and does not provide the absolute number of copies/mL. Instead, the test provides a semi quantitative index (SQI). Lines 115-116 of the original version of the article refers to the ddPCR test, which reports both the wild-type and the mutated copies/mL of plasma as a quantitative PCR test.

  1. Kim, Y.; Shin, S.; Lee, K.-A. A Comparative Study for Detection of EGFR Mutations in Plasma Cell-Free DNA in Korean Clinical Diagnostic Laboratories. Biomed Res. Int. 2018, 2018, 7392419, doi:10.1155/2018/7392419.
  2. Lindeman, N.I.; Cagle, P.T.; Aisner, D.L.; Arcila, M.E.; Beasley, M.B.; Bernicker, E.H.; Colasacco, C.; Dacic, S.; Hirsch, F.R.; Kerr, K.; et al. Updated molecular testing guideline for the selection of lung cancer patients for treatment with targeted tyrosine kinase inhibitors guideline from the college of American pathologists, the international association for the study of lung cancer, and the a. Arch. Pathol. Lab. Med. 2018, 142, 321–346, doi:10.5858/arpa.2017-0388-CP.
  3. Diaz Jr, L.A.; Bardelli, A.; Diaz, L.A.; Bardelli, A. Liquid biopsies: genotyping circulating tumor DNA. J. Clin. Oncol. 2014, 32, 579–586, doi:10.1200/JCO.2012.45.2011.
  4. Roche cobas ® EGFR Mutation Test v2 For in vitro diagnostic use. FDA 2016, 1–71, doi:10.1002/bit.20528.
  5. Keppens, C.; Palma, J.F.; Das, P.M.; Scudder, S.; Wen, W.; Normanno, N.; van Krieken, J.H.; Sacco, A.; Fenizia, F.; Gonzalez de Castro, D.; et al. Detection of EGFR Variants in Plasma: A Multilaboratory Comparison of a Real-Time PCR EGFR Mutation Test in Europe. J. Mol. Diagnostics 2018, 20, 483–494, doi:10.1016/j.jmoldx.2018.03.006.
  6. Wang, L.; Guo, Q.; Yu, W.; Qiao, L.; Zhao, M.; Zhang, C.; Hu, X.; Yang, G.; Xiong, L.; Lou, J. Quantification of plasma EGFR mutations in patients with lung cancers: Comparison of the performance of ARMS-Plus and droplet digital PCR. Lung Cancer 2017, 114, 31–37, doi:10.1016/j.lungcan.2017.10.007.
  7. Pender, A.; Garcia-Murillas, I.; Rana, S.; Cutts, R.J.; Kelly, G.; Fenwick, K.; Kozarewa, I.; Gonzalez de Castro, D.; Bhosle, J.; O’Brien, M.; et al. Efficient Genotyping of KRAS Mutant Non-Small Cell Lung Cancer Using a Multiplexed Droplet Digital PCR Approach. PLoS One 2015, 10, e0139074.
  8. Zhang, Y.; Xu, Y.; Zhong, W.; Zhao, J.; Chen, M.; Zhang, L.; Li, L.; Wang, M. Total DNA input is a crucial determinant of the sensitivity of plasma cell-free DNA EGFR mutation detection using droplet digital PCR. Oncotarget 2017, 8, 5861–5873, doi:10.18632/oncotarget.14390.

This manuscript is a resubmission of an earlier submission. The following is a list of the peer review reports and author responses from that submission.